# Treatment Strategy of Uncontrolled Chronic Rhinosinusitis with Nasal Polyps: A Review of Recent Evidence

**DOI:** 10.3390/ijms24055015

**Published:** 2023-03-06

**Authors:** Sung-Dong Kim, Kyu-Sup Cho

**Affiliations:** Department of Otorhinolaryngology and Biomedical Research Institute, Pusan National University School of Medicine, Pusan National University Hospital, 179 Gudeok-Ro, Seo-gu, Busan 602-739, Republic of Korea

**Keywords:** chronic rhinosinusitis, nasal polyps, endotype, pharmacotherapy, biologics, surgery

## Abstract

Chronic rhinosinusitis (CRS) is recognized as a heterogeneous disease with a wide range of clinical features, resulting in significant morbidity and cost to the healthcare system. While the phenotypic classification is determined by the presence or absence of nasal polyps and comorbidities, the endotype classification has been established based on molecular biomarkers or specific mechanisms. Research on CRS has now developed based on information based on three major endotypes: types 1, 2, and 3. Recently, biological therapies targeting type 2 inflammation have been clinically expanded and may be applied to other inflammatory endotypes in the future. The purpose of this review is to discuss the treatment options according to the type of CRS and summarize recent studies on new therapeutic approaches for patients with uncontrolled CRS with nasal polyps.

## 1. Introduction

Chronic rhinosinusitis (CRS) is a heterogeneous disease with a wide range of clinical features and mechanisms that results in significant morbidity and cost to the healthcare system. CRS affects approximately 3–6% [1,2] of the general population and causes poor quality of life (QOL) and personal productivity in up to 10% of the adult population. CRS leads to more than one million surgical procedures worldwide each year. CRS is diagnosed when there is objective evidence of inflammation or polypoid tissue on endoscopy and CT scans of the sinuses, along with symptoms such as nasal congestion, stuffiness, runny nose, facial pain or tightness, difficulty or loss of smell (anosmia), cough, and fatigue persisting for at least 12 weeks [3].

Although the 2017 European Position Paper on Rhinosinusitis and Nasal Polyposis (EPOS) guidelines divided CRS into two main phenotypes, CRS with nasal polyps (CRSwNP) and CRS without nasal polyps (CRSsNP) [3], the new 2020 EPOS guidelines categorize primary CRS into type 2 and non-type 2 [4]. Type 2 includes allergic fungal rhinosinusitis, eosinophilic CRS among CRSwNP, and central compartment allergic diseases [4]. Recently, as studies on biomarkers reflecting biological mechanisms have been conducted, more personalized medicine has become available.

The EPOS guidelines define CRS as controlled, partially controlled, or uncontrolled, on the basis of objectively determining the degree of subjective symptom reduction, mucosal condition, side effects, need for systemic medications, and need for functional endoscopic sinus surgery.

This review aimed to discuss the treatment options according to the type of CRS and, in particular, to summarize recent studies on new therapeutic approaches for patients with uncontrolled CRSwNP.

## 2. Phenotype and Endotype

The phenotypic classification of CRS is based on clinically observable features, usually distinguished by the presence (CRSwNP) or absence (CRSsNP) of nasal polyps, which are one of the clinically observable features along with discharge in most patients. Patients are typically classified according to the presence or absence of polyps, determine treatment methods, and determine differences in pathological and biological mechanisms. In addition, sub-classification according to comorbidity is used, and asthma and allergy, aspirin-exacerbated respiratory disease (AERD), allergic fungal rhinosinusitis (AFRS), and CRS with immunodeficiency are representative [5,6,7,8,9]. This may be related to the unique features of this subgroup of diseases with high rates of local recurrence, low responsiveness to treatment, and high rates of recurrence after surgical intervention.

CRSsNP is characterized by increased levels of pro-inflammatory cytokines (tumor necrosis factor α and interleukin [IL]-1β), type 1-associated cytokines (interferon [IFN]-γ), transforming growth factor β1, and tissue fibrosis [10,11]. Conversely, CRSwNP has histologically severe eosinophil infiltration and increased IgE and T helper 2 (Th2) cytokines, such as IL-4, IL-5, and IL-13 [10,11]. CRSwNP is a relatively common inflammatory disease in asthma patients, with an increased prevalence of 7–15%, especially in 50% of patients with severe asthma [12,13].

CRSwNP shows different histological characteristics and prevalence depending on the geographical region. European and American patients show high expression of eosinophilia and Th2 cytokines, and Asian patients show more neutrophilic tissue inflammation [14,15]. These different patterns of inflammation suggest that a complex relationship between environmental and genetic factors may contribute to determining predominant expression. In Korea and China, the proportion of eosinophilic nasal polyps has increased significantly with rapid industrialization over the past 20 years [16]. In addition to types classified according to the presence or absence of nasal polyps, in some phenotypes, the characteristics of type 1 and type 2 are combined. Depending on the presence or absence of diseases classified into subgroups, each has unique characteristics that are not included in conventional classifications.

Endotypic classification is based on histological features, such as the presence of neutrophilia, eosinophilia, fibrosis, glandular hypertrophy, and epithelial dysplasia in the early studies [17,18,19]. CRSwNP is associated with eosinophils and mast cells, whereas CRSsNP has a relatively high number of neutrophils [20,21].

Recently, a structured histopathological classification system was proposed based on a combination of effector cell invasion and tissue remodeling changes [22]. Although there is current evidence that polyposis reflects the formation of a fibrin matrix, histopathological features have not been defined in the guidelines and remain experimental [23]. In early molecular studies, IL-5 and IgE were biomarkers for eosinophilic CRS, and IL-8 was a biomarker for neutrophilic CRS [24,25]. Tomassen et al. classified 10 endotypes using cluster analysis for the presence of biomarkers for the differential association between endotypes and the existence of phenotypes of nasal polyps or asthma [2]. The 10 endotypes were subdivided into three subgroups according to low IL-5, absence of IL-5, and high IL-5, including CRSwNP-expressing IgE against Staphylococcus aureus enterotoxins [26]. This study accelerated research on biomarkers by theoretically defining treatment methods according to endotypes. The current view of endotypes can be classified into three endotypes based on their inflammatory profiles, which are composed of specific inflammatory mediators and immune cells observed in tissues [27,28]. Although early studies identified distinct features confined to CD4+ T cells, termed Th1, Th2, and Th17, which contribute to the cytokine patterns observed in type 1, type 2, and type 3 inflammatory endotypes, respectively, other types of cells, including innate lymphoid cells and macrophages, can also be classified into type 1, type 2, and/or type 3 inflammatory profiles (Table 1).

Type 1 inflammatory endotypes are characterized by the expression of T1 cytokines such as IFN-γ and IL-12 produced by Th1 cells, cytotoxic T cells, NK cells, and group 1 innate lymphoid cells (ILC1s). In contrast, type 2 inflammatory endotypes are characterized by the expression of T2 cytokines, such as IL-4, IL-5, and IL-13, produced by Th2 cells, eosinophils, basophils, mast cells, and group 2 innate lymphoid cells (ILC2s). More recently, type 3 inflammatory endotypes have been characterized by the expression of regulatory cytokines such as IL-17 and IL-22 produced by Th17 cells and group 3 innate lymphoid cells (ILC3s).

CRSwNP and CRSsNP are not completely dichotomous depending on the presence or absence of polyps and show overlapping signs of inflammation [29,30,31]. There is also a report that type 2 inflammatory endotypes dominate in both types of phenotypes, and microarray analysis also shows strong similarities between different endotypes [29,30,31,32]. In some CRS patients, a mixture of two or more inflammatory endotypes may appear, and certain cytokines may not be significantly expressed [29].

Several studies have reported that the inflammatory endotype of CRS varies geographically worldwide [31,33,34,35]. In the case of CRSwNP, more than 80% of patients in Western countries show elevated levels of T2 cytokines and eosinophils, which are T2 markers, whereas similar rates of type 2 inflammatory endotype and non-type2 such as type 1 and type 3 are found in Asian patients [31,35].

Recent studies have reported that neutrophilic inflammation may play a role in the pathogenesis of nasal polyps in Asia as well as in the West. In particular, an increase in IL-36 in type 2 CRSwNP and IL-36R+ neutrophils in non-type 2 CRSwNP has been reported in China [36,37,38]. This suggests that the role of neutrophils differs depending on the type and may not be an important biomarker that determines the major inflammatory endotype in nasal polyps.

In the case of CRSsNP, type 1 inflammation associated with elevated IFN-γ is dominant, but other follow-up studies have reported heterogeneous inflammatory patterns [11,33]. Type 2 inflammation was reported in 55% of patients in the United States and 33% and 40% in Europe and Australia [31,39]. Type 1 inflammation, characterized by elevated IFNγ, IL-5, or IL-17, has been reported most frequently in China (58%), but the majority of patients have an atypical phenotype without characteristic increases in cytokines [31].

Therefore, CRSsNP, like CRSwNP, has very heterogeneous and geographically diverse endotypes [31,33,34]. Type 1, type 3, or mixed inflammatory endotypes are common in Asian CRSsNP, and there are reports of CRSsNP with type 2 inflammatory endotypes in the West [31,33,34].

Several studies have reported that genetic and environmental factors can also affect the inflammatory endotypes of CRS. Although 87% of CRSwNP patients in the United States had a type 2 inflammatory endotype, second-generation Asian patients residing in the United States had significantly lower nasal polyp eosinophil counts than non-Asians patients [39,40]. However, longitudinal studies of Asian CRS patients have reported that the prevalence of type 2 inflammatory endotypes has increased over the past 20 years, with a westernized environment being an important factor [41,42].

Furthermore, in China, there are differences according to city. In Beijing, 60% of CRSwNP patients showed type 2 inflammatory endotypes, whereas in Chengdu, it was reported in only 20% of cases, suggesting regional variability even in a single country [31].

This suggests that the CRS endotype may vary with geographical location. Further studies on the effects of genetic or environmental factors on the combination of type 1, type 2, or type 3 inflammatory endotypes are needed.

## 3. Phenotype and Endotype Based Treatment

Although many studies on treatment methods according to phenotype have been reported, reports on the treatment of CRS according to endotype have rarely been performed. This is because the definition of disease recurrence and treatment success is not clear at the CT scan or endoscopy level, and the time points of therapeutic intervention or evaluation are diverse. Despite these limitations, it is necessary to use appropriate pharmacotherapy, surgical approach, or biological agents according to the endotype to prevent repetitive intervention due to recurrence, reduce the risk of disease progression, and tailor treatment for single patients (Table 2).

### 3.1. Antibiotics

According to existing guidelines, antibiotics, such as macrolides and doxycycline, can be considered for the treatment of CRS [3,43]. However, as several studies have reported, there is no recommendation for antibiotic use for CRS, given the lack of placebo-controlled studies [4,44,45]. Bacterial infections based on the endotype are type 3, which is the rationale for antibiotic use in this setting [10]. More than 50% of CRSsNP patient tissues have a partial type 2 endotype, and CRS patients with type 3 endotypes, including those with cost, are likely to respond well to broad-spectrum antibiotics [33]. In a recent study using 625 mg amoxicillin–clavulanic acid, significant objective and subjective results were reported only in the non-type 2 endotype [46]. Although Staphylococcus aureus has been reported to play an important role in type 2 inflammation in CRS, objective studies on the efficacy of antibiotics based on this have been lacking [47,48].

Macrolides are characterized by antibiotic properties and immunomodulation by the inhibition of inflammatory cytokines [16,45,49] and are considered for long-term use as a CRS treatment based on randomized controlled trials [50,51]. Non-type 2 patients with low serum IgE responded well to treatment and showed significantly reduced IL-8 levels [50]. One study targeting eosinophilic CRSwNP patients also showed a significant therapeutic effect; however, additional research is needed to determine whether the effect is greater when used in combination with steroids and biological agents for significant effects in type 1, type 3, or mixed endotypes [52,53,54,55]. Doxycycline is characterized by its antibiotic properties through the inhibition of cytokines and chemokines. It has a significant effect on reducing the size of polyps and improving symptoms by suppressing the type 2 response caused by Staphylococcus aureus in type 2 CRSwNP patients [56,57,58]. Although additional research is needed, it has been reported that the effect of antibiotic response in CRSwNP patients can be predicted using low type 2 biomarkers [59,60].

### 3.2. Corticosteroids

Corticosteroids are considered a mainstream treatment for CRS with anti-inflammatory properties, and are more useful for suppressing type 2 inflammation than type 1 or type 3 inflammation, suppressing ILC2s, Th2 cells, basophils, and eosinophils. This therapy has been used in the treatment of CRSwNP patients rather than CRSsNP [61,62,63,64,65]. Neutrophils are relatively resistant to the effects of corticosteroids, and the reduced effect compared to Western CRSwNP, especially in type 3 inflammation, is supported by studies of Asian CRSwNP and CRSsNP patients [66,67,68,69,70,71,72].

Topical corticosteroid sprays are used in conjunction with oral medications to treat CRS. Although access to sinus tissue is limited, drug delivery has been improved with high-volume steroid irrigation or steroid-impregnated implants [73,74,75,76,77]. A recent study reported that the higher the nasal IL-8 level in CRS patients after surgery, the more difficult it is to control inflammation with topical steroids [78].

In CRS, barrier defects can exacerbate inflammation by increasing the antigen access. Furthermore, epithelial barrier remodeling defects or basal cell hyperplasia induced by type 2 inflammation can be partially ameliorated by corticosteroids [79,80,81,82].

### 3.3. Leukotriene Antagonist

AERD is classified as a type 2 sub-endotype with increased production of prostaglandin D2 (PGD2) and cysteinyl leukotrienes [83]. A recent study reported that PGD2 activates the chemoattractant receptor-homologous molecule expressed on Th2 cells (CRTH2), which is important for the recruitment and activation of eosinophils, basophils, and lymphocytes [84].

### 3.4. Surgery

Surgical treatment can be considered selectively if it does not respond to appropriate pharmacological treatment, and the scope of surgery is controversial [45,85]. Standard endoscopic sinus surgery (ESS) aims to remove inflammatory tissue, ventilate and drain the sinuses, and improve the delivery of topical agents [86]. Although mucus retention due to sinus obstruction promotes microbial overgrowth and infectious inflammation in type 1 and type 3 inflammation, it is relatively less important in CRSwNP and CRSsNP related to type 2 inflammation [87,88].

The important point in standard ESS is to preserve the sinus mucosa as much as possible while extensively dilating the outflow tract of the sinuses and removing the ethmoid lamella, which can cause obstruction [89,90]. The index of successful surgical treatment is the recurrence rate, and the rate of reoperation for 5 years is reportedly 15–20%; the recurrence rate is higher in CRSwNP than in CRSsNP [91]. Although the recurrence rate can be reduced through the use of high-volume corticosteroid nasal irrigation or systemic steroids after surgery, some researchers have reported that more extensive removal of the sinus mucosa, including the floor of the frontal sinus, is required [92,93,94].

Extensive surgery is termed “re-boot” surgery and significant reductions in eosinophilic cationic protein and IL-5 in postoperative nasal secretions have been reported. However, in terms of the endotype, surgical failure is correlated with the presence and intensity of type 2 eosinophilic inflammation and blood eosinophilia [95,96,97,98]. For standard ESS failures, nasalization and Draf III (endoscopic lothrop) can be considered, including re-boot surgery [92,93,94]. The nasalization procedure aggressively removes the middle turbinate and paranasal mucosa to induce healing of normal mucosa. The Draf III procedure maximizes access to the frontal and ethmoid sinuses by removing all the bones and mucosa of the upper part of the middle turbinate and the floor of the frontal sinuses. Re-boot surgery includes both the previous surgeries to remove mucosa in the nasal and paranasal sinuses and the floor of the frontal sinuses [92,93,94]. Although the effectiveness of the aggressive surgical approach is controversial, there are reports of positive results in recurrent or high-risk type 2 type CRSwNP [95]. In addition, there is a potential benefit of improved drug delivery after surgical treatment, and some studies have reported that it is effective in type 1 or type 3 inflammatory endotypes [97].

Several studies have reported on the prediction of surgical outcomes according to endotype. In one study, cluster analysis was performed to determine the correlation between endotypes and treatment outcomes. The treatment outcome was worst when asthma was accompanied by type 2 inflammation, in which IL-5, Immunoglobulin E (IgE), and eosinophils were increased in the nasal mucosa [99]. In another study, the T2 cytokine IL-5 and the type 2 biomarkers periostin and C-C motif chemokine ligand 26 (CCL26) were higher in patients with difficult-to-treat CRSwNP but were not associated with treatment outcome. However, yet another study reported that type 2 inflammation correlated with the epithelial secreted cysteine proteinase inhibitor cystatin SN and was associated with poor outcome [100]. Some studies have reported that higher intensities of type 1 and type 3 inflammations are associated with surgical failure [99,101].

### 3.5. Biological Treatment

Biologics, a new treatment option for CRS, are increasingly being used mainly in CRSwNP patients and are effective in suppressing type 2 inflammation and minimizing side effects. Several monoclonal antibodies have been approved or are under development for the treatment of CRS, particularly in association with asthma, secondary or exacerbated disease, and recurrence after surgery (Table 3).

Omalizumab is a humanized recombinant DNA-derived IgG1k monoclonal antibody that specifically binds to free IgE in interstitial and blood fluid and to membrane-bound forms of IgE (mIgE) on the surface of mIgE-expressing B lymphocytes [102]. The mechanism of reduced sensitivity to allergen stimulation is the gradual downregulation of FcεRI receptors on basophils, mast cells, and dendritic cells by free IgE binding to omalizumab. In particular, omalizumab does not bind to IgE, which is already bound to the high-affinity IgE receptor (FceRI) on the surface of antigen-presenting dendritic cells, basophils, and mast cells [103]. Nasal polyps, local eosinophilic inflammation, and asthma are associated with elevated local IgE levels. Several studies have reported that omalizumab reduces the size of polyps and significantly improves the quality of life of CRSwNP patients [104,105,106]. Although endoscopic finding and CT score showed improvement, there was no significant improvement in tissue and blood eosinophilia [107]. Based on this, Omalizumab was approved by the FDA for the treatment of nasal polyps in 2020.

Reslizumab and Mepolizumab are monoclonal antibodies that block IL-5, which contributes to the activation, maturation, and survival of eosinophils [108]. IL-5 is an important mediator of tissue eosinophilia in CRSwNP patients and is derived from T cells and ILC2s [109]. Reslizumab reported improvement in nasal polyp scores in a small study, and mepolizumab reported significant improvements in nasal polyp severity and symptom scores, along with a reduction in the need for surgery in CRSwNP patients in a multicenter randomized controlled trial study [110,111]. In particular, Mepolizumab showed a significant improvement to visual analogue score (VAS), Sino-Nasal Outcome Test (SNOT-22) score, nasal polyposis severity, and endoscopic nasal polyp score compared with placebo groups [112]. Based on this, mepolizumab has recently been reported in phase 3 trials for subjective and objective efficacy for CRSwNP, and the FDA approved its use [112].

Benralizumab is a cytotoxic monoclonal antibody that blocks the IL-5 receptors to eliminate eosinophils. Several studies have reported that it reduces annual asthma exacerbation rates and increases the lung function in uncontrolled asthma. Moreover, a phase 3 trial of CRSwNP was conducted [113,114,115,116]. However, additional studies on CRS patients are needed in the future.

Dupilumab is a monoclonal antibody that blocks IL-4 and IL-13, which are important in type 2 inflammation by binding to the α component of the shared receptor and is approved for the treatment of unregulated CRSwNPs. In two multicenter phase 3 trial randomized controlled trials, dupilumab improved quality of life in patients with severe CRSwNP, as well as reported reductions in nasal polyp size, nasal congestion, and sense of smell [117]. In addition, Ryu et al. reported that dupilumab significantly reduced the use of systemic corticosteroids compared to placebo and reduced nasal surgery by 76%. When serum and nasal secretions of CRSwNP patients were analyzed, type 2 inflammatory markers were significantly reduced in nasal secretions [118]. Table 4 summarizes the effects of FDA approved biologics for the treatment of CRS.

In a study comparing dupilumab and functional endoscopic sinus surgery (FESS) in CRSwNP patients, both methods were effective in reducing symptoms following the Sino-Nasal Outcome Test (SNOT-22). Furthermore, patients treated with FESS reported greater reductions in polyp burden than patients treated with dupilumab, whereas patients treated with dupilumab reported an improved sense of smell and greater reductions in postnasal drip, cough, and thick nasal drainage [119]. When the combination use of FESS and biologics is required, a retrospective matched cohort study on the timing of biologic use was reported [120]. The treatment effect size of dupilumab versus placebo was generally significantly greater in patients who had recent surgery, especially within 3 years. On the other hand, the number of surgeries did not show significant results.

Different biologics and ASA-D reported significant clinical improvement in patient outcomes in a meta study including 29 randomized controlled trials evaluating eight treatments (*n* = 3461) comparing monoclonal antibodies and aspirin desensitization (ASA-D) in CRSwNP patients [121]. In this study, dupilumab showed the best results for all indicators, including sinusitis symptoms, quality of life, and rescue of oral corticosteroids.

Although monoclonal antibodies are effective drugs for suppressing type 2 inflammation, there are no guidelines on which biological agents should be used first in patients with type 2 inflammation, because each target mechanism is different. There are reports of the various effects mentioned above, but a direct comparative evaluation for each has not been conducted. However, several recommendations for clinicians were presented at a recent expert board meeting, and dupilumab was reported to have the highest efficacy and objectivity among the currently available monoclonal antibodies [122]. However, a dosing interval has not yet been established. In a randomized, double-blind, phase 3 trial of dupilumab, there was no statistically significant difference in efficacy between a group of patients treated with dupilumab every 2 weeks for 52 weeks and another group treated with dupilumab every 2 weeks for 24 out of 52 weeks and then every 4 weeks for 28 weeks [117]. This may be positive in terms of patient burden if a certain level of therapeutic effect can be maintained by extending the dosage interval.

The probability of reoperation within 5 years is reported to be 20%, but it is also questionable whether biologics should be used first in patients who have not undergone surgery [123,124]. Although these drugs are very effective, there is no change in the quality of life compared to before treatment, and nasal polyps may not be cured [117].

The ideal treatment strategy should be determined for each patient; however, in the case of patients with multiple recurrences, the combination of biological agents after planned surgery may be the most effective and practical treatment.

## 4. Conclusions

CRS was previously classified based on its phenotype; however, as the knowledge of immunopathological mechanisms has significantly increased, it is necessary to establish a new pathway for the treatment of CRS patients based on the three major endotypes of type 1, type 2, and type 3.

CRS may be differentiated between other types and type 2 inflammation of immune responses associated with more severe clinical features and comorbidities, such as relapse and asthma. Type 2 inflammation can be found in both the CRSwNP and CRSsNP phenotypes, and is higher in the West than in Asia, but is also increasing in Asia due to rapid industrialization.

Although many studies have demonstrated the safety and efficacy of FDA-approved biologics are being reported, there are currently no guidelines for the selection of specific biologics for individual patients. To prevent inefficient treatment and excessive burden on patients, regular clinical monitoring of biological agents and the establishment of appropriate durations and intervals of use are also required.

## Figures and Tables

**Table 1 ijms-24-05015-t001:** Characteristics of inflammatory endotype in CRS.

Endotype	T1	T2	T3
**Origin cell**	Th1 cell ILC1	Th2 cell ILC2	Th17 cell ILC3
**Inflammatory cytokines**	TNF-α, IL-1β, IFN-γ, IL-12	IL-4, IL-5, IL-13	IL-17
**Effector cells**	NK cell M1 Macrophage	Basophil Eosinophil M2 Macrophage	Neutrophil
**Target pathogens**	Microbes, protozoa, viruses	Parasite	Bacteria, fungi
**Clinical characteristics**	Mucopurulent discharge Nasal polyp (limited)	Anosmia, Nasal polyposis, Asthma	Mucopurulent discharge Nasal polyp (limited)

Th, T helper cell; ILC, innate lymphoid cell; TNF-α, tumor necrosis factor; IFN-γ, interferon gamma; IL, interleukin; NK, natural killer.

**Table 2 ijms-24-05015-t002:** Potential management of inflammatory endotype in CRS.

Endotype	T2	Non-T2
**Pharmacotherapy**	Topical and oral corticosteroids Doxycycline	Macrolide
**Surgery**	Nasalization Endoscopic Lothrop Re-boot surgery	Conventional FESS
**Biologics**	Omalizumab Mepolizumab Dupliumab Benralizumab Reslizumab	Not defined yet

FESS, functional endoscopic sinus surgery.

**Table 3 ijms-24-05015-t003:** Biologics for the target of CRS and approval status.

Biologics	Trade Name	Target	Approval State
**Omalizumab**	Xolair^®^	anti IgE	FDA approved
**Mepolizumab**	Nucala^®^	anti IL-5	FDA approved
**Dupilumab**	Dupixent^®^	anti IL-4Rα	FDA approved
**Benralizumab**	Fasenra^®^	anti IL-5Rα	Phase 3 trials (FDA approved for severe asthma)
**Reslizumab**	CINQAIR^®^	anti IL-5	Phase 2 trials (FDA approved for severe asthma)

Ig E, immunoglobulin E; IL, interleukin.

**Table 4 ijms-24-05015-t004:** Effects of biologics on CRS.

Biologics	Increase of Quality of Life and Olfaction	Endoscopic Nasal Polyp Score	Lund-Mackay CT Scan Score	Reduction of Serum IgE Levels	Reduction of Tissue Eosinophil Levels	Reduction of Blood Eosinophil Levels
**Omalizumab**	O	O	O	O		
**Mepolizumab**	O	O			O	O
**Dupilumab**	O	O	O	O	O	O

## Data Availability

The study did not report any date.

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
