# Peer review of "Treatment Strategy of Uncontrolled Chronic Rhinosinusitis with Nasal Polyps: A Review of Recent Evidence"

_ijms, 2023, doi:10.3390/ijms24055015_

Round 1

Reviewer 1 Report

The topic is very interesting from a practical point of view but, being a review, I think you should modify it a lot before publishing

MAJOR COMMENT

1) First of all I suggest you to have the article reviewed by an english native speaker, because many sentences are not clear.

2)The definition of uncontrolled CRS is lacking

3) The paper is a review about therapy of CRS, but there is a disproportion between the part concerning CRS and the one concerning the therapy. The first part concerning CRS in general is exaustive, while the part of therapy is poor.  

In the part dedicated to therapy,  you should describe with more details ALL the articles available about the efficacy of  the single (antiIge, anti IL5...) monoclonal antibodies on CRS.

You can also insert a Summary Table

E.G.: On anti IgE you should describe all articles about this therapy, its efficacy on symptoms and objective parameters of CRS etc. as you did for Dupilumab. In conclusion you should try to indicate which patients will benefit from this therapy.

( There are many papers about this topic written by Bachert et al.. Canonica et al ect... )

Finally, from a practical point of view, after how many surgical interventions do you suggest the use of monoclonal antibodies?  

MINOR COMMENT

Raw 36 This what? (CRS?)

RaW 40 persisting

Raw 46 it is available a more personalized medicine

Raw 49 to summarize

Raw 54 to 61 Modify!!  english not clear

raw 65 delete expression

Raw 75 Delete particularly

Raw 79-81 Modify!! Not clear

Raw 84 references 15-17??? 

Raw 129-131 Not clear .Specify the role of IL36 

Raw 147 Add  On the contrary  longitudinal studies

Raw 79-81 Modify!! Not clear 

Raw 129-131 Not clear 

Raw 147 Add  On the contrary  longitudinal studies ( there are two references!!)

Raw 165 for the single patient

Raw 170 delete oral and topical corticosteroids

Raw 172 avoidance of overuse of antibiotics????

Raw 173-176 Modify Not clear!!

Raw 184 showed instead of represented

Raw 198 this therapy has been used

Raw 203 has been improved

Raw 215-216 It is thus???

Raw 246 effective instead of significant

Raw 255-256  Not clear rewrite

Raw 263 if associated with asthma

Raw 267-270 Rewrite English  not clear!! 

Raw 273 several studies: how many???

RaW 285 approved its use

Raw 287-291 DELETE you are speaking about CRS..not asthma!!

Raw  294 Inappropriately regulated CRSwNPs???

raw 312 In sthis tudy instead of  also

raw 333  combination of treatment? What do you mean? Specify better with references 

Raw 337 classified instead of understood

Raw 346-7 Modify the sentence  english not clear 

REFERENCES

54 is lacking in the text

83 is the same of 43

Author Response

Reviewer #1:

MAJOR COMMENT

Comment 1: First of all I suggest you to have the article reviewed by an english native speaker, because many sentences are not clear.

Response 1: As you pointed out, we received English proofreading.

Comment 2: The definition of uncontrolled CRS is lacking

Response 2: As you pointed out, we additionally described the definition of uncontrolled CRS in INTRODUCTION.

Comment 3: The paper is a review about therapy of CRS, but there is a disproportion between the part concerning CRS and the one concerning the therapy. The first part concerning CRS in general is exaustive, while the part of therapy is poor.  

Response 3: As you pointed out, the contents of the part of therapy were additionally described, and a table was also added.

Comment 4: In the part dedicated to therapy, you should describe with more details ALL the articles available about the efficacy of the single (antiIge, anti IL5...) monoclonal antibodies on CRS.

Response 4: As you pointed out, the effect of the monoclonal antibody and the timing of its use when combined with surgical treatment were additionally described.

Comment 5: You can also insert a Summary Table. E.G.: On anti IgE you should describe all articles about this therapy, its efficacy on symptoms and objective parameters of CRS etc. as you did for Dupilumab. In conclusion you should try to indicate which patients will benefit from this therapy. ( There are many papers about this topic written by Bachert et al.. Canonica et al ect..)

Response 5: As you pointed out, table 4 summarizing the effects of biologics on CRS was added. In the table, the effect according to the objective and subjective parameters of each FDA approved biologics were summarized.

Comment 6: Finally, from a practical point of view, after how many surgical interventions do you suggest the use of monoclonal antibodies?  

Response 6: A retrospective matched cohort study on the timing of biologic use was reported when the combination use of FESS and biologics is required. The treatment effect size of dupilumab versus placebo was generally significantly greater in patients who had recent surgery, especially within 3 years. On the other hand, the number of surgeries did not show significant results. Therefore, in order to increase the treatment effect, it is more important to use biologics within 3 years after surgery rather than the number of surgeries, and this was added to the manuscript.

MINOR COMMENT

Response 1: As you pointed out, we corrected minor comments and received English proofreading.

Reviewer 2 Report

I do not agree the summary in Table 1. Especially effector cells and pathogens items. No mention is made about IL-36 in the table.

L135: Type 1 inflammation characterized by elevated IFNγ, IL-5, or IL-17 has been reported most frequently in China,

The authors had better show the percentage values.

L166: Table 2. Potential management of inflammatory endotype in CRS.

Erratum in Surgery items?

L308: Multiple biologics and ASA-D reported significant clinical improvement in patient outcome in a meta study.

No reference is cited.

The references are not aligned according to the instructions.

There are countable sites of inadequate use of abbreviations, grammar errors, erratum in the manuscript, and awkward phrasings that detract from this work as listed below.

Eg. L23: “CRS has now developed based on information based on three major endotypes”

Author Response

Reviewer #2:

Thank you for careful review and kind revision of our manuscript.

Comment 1: I do not agree the summary in Table 1. Especially effector cells and pathogens items. No mention is made about IL-36 in the table.

Response 1: The role of neutrophils varies by type and may not be an important biomarker for determining the major inflammatory endotypes in nasal polyps. Studies on the relationship between type 2 CRSwNP and neutrophilic inflammation have been reported, but the role could not be included in the table because additional research is needed and the evidence is not sufficient. We added the revised contents to the manuscript.

Comment 2: L135: Type 1 inflammation characterized by elevated IFNγ, IL-5, or IL-17 has been reported most frequently in China, The authors had better show the percentage values.

Response 2: As you pointed out, Percentage values ​​have been added to the manuscript.

Comment 3: L166: Table 2. Potential management of inflammatory endotype in CRS. Erratum in Surgery items?

Response 3: As you pointed out, we modified the table.

Comment 4: L308: Multiple biologics and ASA-D reported significant clinical improvement in patient outcome in a meta study. No reference is cited.

Response 4: As you pointed out, we cited references

Comment 5: The references are not aligned according to the instructions. There are countable sites of inadequate use of abbreviations, grammar errors, erratum in the manuscript, and awkward phrasings that detract from this work as listed below.

Response 5: As you pointed out, we received English proofreading.

Reviewer 3 Report

Good morning, the paper submitted for evaluation is a typical Review that includes a large number of papers from previous years on chronic rhinosinusitis.

As a reviewer, I indicate points for improvement: 1. the title should include the plural, i.e. "... nasal Polyps..." 2. line 23 needs correction (using "based on" twice is confusing) 3. line 28 use of plural (polyps) 4. line 41 abbreviation EPOS needs to be expanded 5. lines 52-55 need correction (use of "clinically observable" several times causes confusion) 6. line 211 abbreviation AERD needs to be expanded 7. line 259 - "Biological treatment" should be numbered 3.5 8. line 336 - "4. Conclusion" requires the use of a large and bold font Regards.

Author Response

Reviewer #3:

Thank you for careful review and kind revision of our manuscript.

Good morning, the paper submitted for evaluation is a typical Review that includes a large number of papers from previous years on chronic rhinosinusitis.

As a reviewer, I indicate points for improvement:

Comment 1: the title should include the plural, i.e. "... nasal Polyps..."

Response 1: As you pointed out, we have corrected the title.

Comment 2: line 23 needs correction (using "based on" twice is confusing)

Response 2: As you pointed out, we have corrected the sentence.

Comment 3: line 28 use of plural (polyps)

Response 3: As you pointed out, we have corrected the sentence.

Comment 4: line 41 abbreviation EPOS needs to be expanded

Response 4: As you pointed out, we have corrected the sentence.

Comment 5: lines 52-55 need correction (use of "clinically observable" several times causes confusion)

Response 5: As you pointed out, we have corrected the sentence.

Comment 6: line 211 abbreviation AERD needs to be expanded

Response 6: AERD is first expanded mentioned in 2. Phenotype and Endotype.

Comment 7: line 259 - "Biological treatment" should be numbered 3.5

Response 7: As you pointed out, we corrected the number.

Comment 8: line 336 - "4. Conclusion" requires the use of a large and bold font Regards.

Response 8: As you pointed out, we corrected the word.

Reviewer 4 Report

A well performed review on the phenotype and endotype of CRS. There is lack of mention of the allergic phenotype of CRS. This would be a useful additional information to give a comprehensive picture of the management of CRS.

Some of these references could be added:

1: Abdullah B, Vengathajalam S, Md Daud MK, Wan Mohammad Z, Hamizan A, Husain S.

The Clinical and Radiological Characterizations of the Allergic Phenotype of

Chronic Rhinosinusitis with Nasal Polyps. J Asthma Allergy. 2020 Oct

27;13:523-531. doi: 10.2147/JAA.S275536. PMID: 33149624; PMCID: PMC7602905.

2: Delgado-Dolset MI, Obeso D, Sánchez-Solares J, Mera-Berriatua L, Fernández P,

Barbas C, Fresnillo M, Chivato T, Barber D, Escribese MM, Villaseñor A.

Understanding Systemic and Local Inflammation Induced by Nasal Polyposis: Role

of the Allergic Phenotype. Front Mol Biosci. 2021 May 14;8:662792. doi:

10.3389/fmolb.2021.662792. PMID: 34055883; PMCID: PMC8160224.

Author Response

Reviewer #4:

Thank you for careful review and kind revision of our manuscript.

A well performed review on the phenotype and endotype of CRS. There is lack of mention of the allergic phenotype of CRS. This would be a useful additional information to give a comprehensive picture of the management of CRS.

Comment 1: Some of these references could be added:

1: Abdullah B, Vengathajalam S, Md Daud MK, Wan Mohammad Z, Hamizan A, Husain S. The Clinical and Radiological Characterizations of the Allergic Phenotype of Chronic Rhinosinusitis with Nasal Polyps. J Asthma Allergy. 2020 Oct 27;13:523-531. doi: 10.2147/JAA.S275536. PMID: 33149624; PMCID: PMC7602905.

2: Delgado-Dolset MI, Obeso D, Sánchez-Solares J, Mera-Berriatua L, Fernández P, Barbas C, Fresnillo M, Chivato T, Barber D, Escribese MM, Villaseñor A. Understanding Systemic and Local Inflammation Induced by Nasal Polyposis: Role of the Allergic Phenotype. Front Mol Biosci. 2021 May 14;8:662792. doi:10.3389/fmolb.2021.662792. PMID: 34055883; PMCID: PMC8160224.

Response 1: As you pointed out, we added 2 references.

Round 2

Reviewer 1 Report

You modify the article, as I suggested.

Only minor comment before publishing

It should be interstenting if you write in the text how many paper ( NUMBER) are now available  regarding  the use of each monoclonal antibodies in CRS . I asked you before about antiIgE

Raw 219 ?????

raw 291 showed instead of was

raw 325 different instead of multiple

Author Response

Thank you for careful review and kind revision of our manuscript.

Reviewer #1:

You modify the article, as I suggested.

Only minor comment before publishing

Comment 1: It should be interesting if you write in the text how many paper ( NUMBER) are now available  regarding  the use of each monoclonal antibodies in CRS . I asked you before about antiIgE

Response 1: Studies on the clinical effects or mechanism of monoclonal antibodies are continuously being published, but the number of papers that serve as the basis for FDA approval in CRS is limited. The RCT studies cited in this paper could be important for clinical application.

The most representative biologics targeting anti-IgE is omalizumab, which has a trade name of Xolair®. As you pointed out before, we added information on omalizumab’s clinical efficacy on symptoms and objective parameters of CRS (quality of life and olfaction, endoscopic nasal polyp score, Lund-Mackay CT score, serum IgE level, tissue eosinophil level and blood eosinophil level), and based on this, Table 4 was prepared.

Comment 2: Raw 219 ?????

Response 2: The sentence you pointed out was deleted because it was unnecessary.

Comment 3: raw 291 showed instead of was

Response 3: As you pointed out, we corrected the sentence.

Comment 4: raw 325 different instead of multiple

Response 4: As you pointed out, we corrected the sentence.

Reviewer 3 Report

Dear Authors, thank you for correcting the work - all my comments have been taken into account. I wish you good luck in the further stages of publishing your article. Regards.

Author Response

Thank you for careful review and kind revision of our manuscript.